# ACTIVATION-AWARE PRUNING OF LARGE LANGUAGE MODELS

## ABSTRACT

Although large language models (LLMs) have performed well across various tasks since emergence, their application in many specific scenarios is hindered by limited computational resources. One-shot pruning mitigates this issue by removing redundant parameters from the weight matrix in a single training run. However, most existing approaches still depend on heuristic searches or linear approximations inherited from deep networks, thereby assigning equal importance to all weight matrices while overlooking the activation-function modules in Transformer architectures—modules that alter the relative significance of weights before and after activation. In this paper, we propose a novel pruning method, Activation-Aware Pruning (*AAP*), which improves compression performance by explicitly capturing the shifts induced by activation. Beyond solely matching pre-activation outputs, *AAP* incorporates activation-aware regularization that preserves post-activation sign and pattern consistency, substantially reducing accuracy degradation at high sparsity levels. Moreover, we propose an approximate update rule based on an analytical approximation of the weight matrix, which requires no fine-tuning and is supported by theoretical guarantees. Applied to the open-source models, e.g., OPT model family and LLaMA series, our method achieves lower perplexities at different sparsities compared with prior approaches. The code will be released soon on GitHub.

## 1 INTRODUCTION

The emergence of large language models (LLMs) has significantly advanced the field of natural language processing. From text classification and sentiment analysis to natural language understanding and text generation, LLMs exhibit advantages far beyond previous algorithms and models. As LLMs continue to develop, their parameter sizes have expanded rapidly, reaching scales as large as hundreds of billions, such as the OPT-175B. Consequently, running these large models requires massive storage and computational resources, resulting in LLMs being difficult to deploy in certain environments.

To address LLM compression, early works combined pruning with quantization and coding Han et al. (2015); Molchanov et al. (2016) or used iterative retraining to recover accuracy Luo et al. (2017); Liu et al. (2018), albeit with high cost. One-shot methods He et al. (2017); Singh & Alistarh (2020) reduced overhead by pruning pre-trained models without retraining, and optimization-based techniques Frantar & Alistarh (2022); Wang et al. (2019) mitigated Hessian estimation cost. Yet most still rely on heuristics, for example, SparseGPT Frantar & Alistarh (2023) approximates OBS (Optimal Brain Surgeon) Hassibi & Stork (1992) via partial updates, Wanda Sun et al. (2023) scores by magnitude–activation products, and Zhang et al. (2023) iteratively updates masks. More principled approaches like ADMM-based LLM Pruning in one-Shot (ALPS) Meng et al. (2024a) cast pruning as a constrained optimization via operator splitting, improving precision at high sparsity. Recently, several studies have leveraged activation information to guide pruning, such as Zhao et al. (2022); Wang et al. (2025) and Activation-aware Weight Quantization (AWQ) Lin et al. (2024), which select channels based on activation statistics, but they still lack solid theoretical grounding.

All these works have focused solely on exploring the pre-activation loss or post-activation information. Although they perform well at moderate sparsity levels, their performance often degrades significantly under high sparsity requirements. We contend that the core limitation is that, although

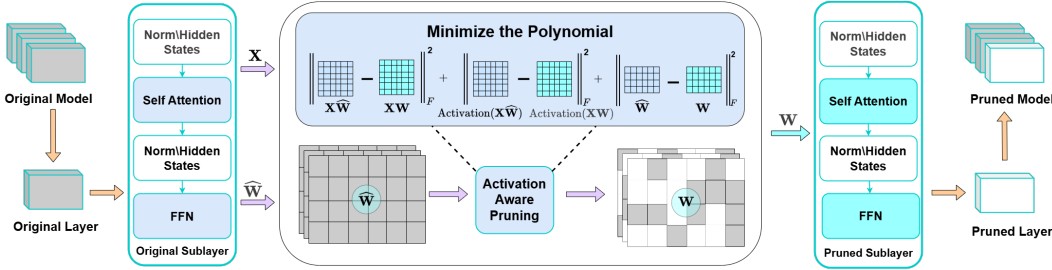

Figure 1: Overview of the proposed *AAP* algorithm. By adding activation-aware regularization (in the center of the loss polynomial) to capture both pre-activation and post-activation information, we can achieve a better performance. $X$ represents the input of each selected sublayer, $\hat{W}$ represents the original sublayer weight matrix, $W$ represents the pruned sublayer weight matrix. Our work primarily focuses on pruning the self-attention and FFN sublayers.

activation mechanisms have become indispensable to modern architectures (e.g., CNNs, Transformers), existing pruning methods rarely capture—or explicitly account for—their influence. This lack of attention to the changes in information before and after activation causes these methods to lose focus in large-scale sparse compression, often leading to suboptimal pruning outcomes.

In this paper, we present a one-shot pruning framework Activation-Aware Pruning (*AAP*) for LLMs that combines numerical optimization with an activation-aware optimization. The overall process is illustrated in Figure 1. At the core of our method is activation-aware regularization which explicitly quantifies how the significance of each weight matrix $W$ evolves from pre-activation to post-activation. Incorporating this module reduces Hessian information loss and delivers consistently better perplexity at a fixed sparsity level. Thus, building on Meng et al. (2024a), with the additional post-activation consistent loss, we re-derive the iterative scheme, resulting in an algorithm that explicitly accounts for both pre-activation and post-activation information. By combining the operator-splitting technique with the augmented Lagrangian approach, we derive a unified update that adjusts $\mathbf{W}$ only once while jointly accounting for both pre-activation and post-activation signals. Owing to its Lipschitz property, the resulting pruning rule is inherently robust to external perturbations, fully self-contained, and straightforward to integrate into production pipelines, greatly simplifying downstream deployment. Through mathematical approximation and update algorithm, we generate a highly sparse weight matrix $\mathbf{W}$ for each selected sublayer and insert the resulting weights back into the original layer, thereby achieving effective layer compression. The compressed layers are immediately put into use, eliminating the need for retraining or fine-tuning, which significantly reduces both time and computational cost. In addition, by independently pruning each sublayer weight matrix, *AAP* preserves sufficient flexibility for future parallelization and scalability upgrades.

Our main contributions are summarized as follows:

- **Activation-aware:** We propose a novel activation-aware regularization approach that aligns the pre-activation and post-activation importance, thereby reducing information loss during pruning. We conduct targeted pruning experiments on selected sublayers to demonstrate how the activation term sensitively captures changes in weight significance and improves the performance of each weight matrix.

- **Pruning algorithm and Theory analysis:** We propose an activation-aware pruning algorithm that leverages the Lipschitz property of the activation function. Theoretical analysis guarantees that activation-aware regularization preserves the iterative updates, while the approximate weight updates closely match the target matrices, thereby ensuring effective model pruning.

- **Better performance:** Our results highlight that *AAP* bridges the long-standing gap between accuracy and efficiency in large-scale pruning. By jointly preserving pre-activation and post-activation signals, *AAP* consistently achieves state-of-the-art sparsity–accuracy trade-offs, yielding up to 12% relative improvements over strong baselines such as ALPS and substantially outperforming heuristic methods. Importantly, its benefits amplify at extreme sparsity,

where conventional approaches collapse, yet *AAP* sustains robust performance across diverse corpora. These properties make *AAP* not only an effective pruning strategy but also a practical enabler for deploying multi-billion-parameter LLMs under real-world resource constraints.

## 2 RELATED WORKS

### 2.1 ONE-SHOT PRUNING

Early one-shot pruning methods Mozer & Smolensky (1989); Guo et al. (2016); Han et al. (2015) such as Single-shot network pruning (SNIP) Lee et al. (2018), GraSP Wang et al. (2020), and EP Tanaka et al. (2020) assess weight importance at initialization and heuristically remove less sensitive weights, inspiring later approaches like Wanda Sun et al. (2023). While effective at reducing parameters, these heuristics often miss globally optimal sparsity patterns. Subsequent works sought stronger theoretical grounding: Chen et al. (2021) built inherently sparse networks from training start; OBS-based methods Hassibi & Stork (1992); Frantar & Alistarh (2023) improved interpretability via block-wise inverse updates; and Meng et al. (2024b) used local combinatorial optimization for structured pruning. This shift toward principled methods culminated in inference-aware ZipLM Kurtic et al. (2023) and ALPS Meng et al. (2024a), which frames pruning as a quadratic program with activation constraints solved by ADMM and PCG Boyd et al. (2011); Davis & Yin (2016); Nocedal & Wright (1999), achieving superior sparsity–accuracy trade-offs, especially in preserving pre-activation information.

### 2.2 PARAMETER-EFFICIENT TRANSFER LEARNING

Parameter-efficient transfer learning (PETL) is a deep learning paradigm designed to reduce the number of parameters that need to be updated during transfer learning, making it particularly suitable for scenarios with limited computational resources. It can be applied to large pre-trained models such as BERT Devlin (2018) and RoBERTa Liu (2019). Early approaches, such as Adapter modules Houlsby et al. (2019); Pfeiffer et al. (2020), insert small bottleneck layers into each Transformer block and train only these newly introduced modules. Building on this idea, LoRA Hu et al. (2021) introduces low-rank adaptation, which freezes the original pre-trained weights and injects trainable low-rank decomposition matrices into each Transformer layer, thereby significantly reducing the number of trainable parameters required for downstream tasks. Meanwhile, prompt-based and prefix-based tuning methods Li & Liang (2021); Lester et al. (2021) further minimize parameter overhead by optimizing soft prompts either at the embedding level or at intermediate Transformer layers. More recently, UniAdapter Lu et al. (2023) unifies unimodal and multimodal adapters, enabling parameter-efficient cross-modal adaptation for pre-trained vision-language models.

### 2.3 ACTIVATION-INFORMATION GUIDED COMPRESSION

Recent approaches leverage activation information in various ways: attention-energy masks Zhao et al. (2022), low-rank or second-order activation criteria Yuan et al. (2025); Mi et al. (2025), and CFSP's single-pass sparsity allocation Wang et al. (2025) use activation statistics to guide pruning, while dynamic methods such as uMoE Koike-Akino et al. (2025) and DuoGPT Yin et al. (2025) adjust weights at inference, and AWQ Lin et al. (2024) preserves a subset of high-activation channels for quantization. In contrast, *AAP* integrates both weight similarity and activation consistency into a single framework, avoiding reliance on specific activation distributions or dynamic control. This design generalizes more robustly across datasets and architectures, scales naturally to billion-parameter LLMs, and introduces no additional inference overhead. Empirically, *AAP* delivers consistently larger perplexity reductions, highlighting its stronger effectiveness and practicality compared with specialized or task-dependent activation-based methods.

Building on previous works, we propose *AAP*, which captures activation-induced shifts in the importance of $W$ and integrates this information into the existing numerical optimization framework. This unified update drives the solution closer to the global optimum, avoiding suboptimal local minima.

## 3 METHOD

During the post-training pruning of large language models (LLMs), the full-model compression problem is decomposed into a series of layer-wise subproblems. The objective of pruning is to identify a weight matrix that minimizes the reconstruction error between the outputs of the original and pruned layers, subject to a specified constraint. In this section, we consider the one-shot effective LLMs pruning under Activation-Aware Pruning. Furthermore, we provide a theoretical analysis to support the validity and effectiveness of the proposed method.

### 3.1 ONE-SHOT PRUNING UNDER GENERAL CONSTRAINT

For each layer-wise subproblem, let $\widehat{\mathbf{W}} \in \mathbb{R}^{N_{\text{in}} \times N_{\text{out}}}$ be the weight matrix of the layer, where $N_{\text{in}}$ and $N_{\text{out}}$ denote the input and output dimension for such layer. Assume there are $N$ samples and $L$ sequence length, the input for such layer is represented as $\mathbf{X} \in \mathbb{R}^{NL \times N_{\text{in}}}$. Then, the aim to prune the weight matrix $\widehat{\mathbf{W}}$ is to find the alternative weight matrix $\mathbf{W}$ to approximate the original weight matrix. The common approach is to define the loss functions that minimize 1) the reconstruction error between $\mathbf{W}$ and $\widehat{\mathbf{W}}$ Kurtic et al. (2023); and 2) the reconstruction error between $\mathbf{X}\mathbf{W}$ and $\mathbf{X}\widehat{\mathbf{W}}$ Meng et al. (2024b); 3) both reconstruction errors Meng et al. (2024a), which prevents $\mathbf{W}$ from diverging too far from the original weights.

Formally, the one-shot pruning problem can be formulated as following

$$\min_{\mathbf{W}, \mathbf{D} \in \mathbb{R}^{N_{\text{in}} \times N_{\text{out}}}} \|\mathbf{X}\widehat{\mathbf{W}} - \mathbf{X}\mathbf{W}\|_F^2 + \lambda_2 \|\widehat{\mathbf{W}} - \mathbf{W}\|_F^2,$$

$$\text{s.t. } R(\mathbf{W}) \leq 0,$$

where $R(\mathbf{W})$ is the constraint function, such as $l_0$-constrained function and low-rank constrained function. In order to solve the optimization problem, the typical method is combining the operator-splitting technique and augmented Lagrangian approach. In particular, the additional matrix $\mathbf{D}$ is introduced to approximate the matrix $\mathbf{W}$. Then, the original objective is reformulated as:

$$\min_{\mathbf{W}, \mathbf{D} \in \mathbb{R}^{N_{\text{in}} \times N_{\text{out}}}} \|\mathbf{X}\widehat{\mathbf{W}} - \mathbf{X}\mathbf{W}\|_F^2 + \lambda_2 \|\widehat{\mathbf{W}} - \mathbf{W}\|_F^2 + R(\mathbf{D}),$$

$$\text{s.t. } \mathbf{W} = \mathbf{D},$$

The augmented Lagrangian function of the objective function is as following:

$$L_\rho(\mathbf{W}, \mathbf{D}, \mathbf{V}) = \|\mathbf{X}\widehat{\mathbf{W}} - \mathbf{X}\mathbf{W}\|_F^2 + \lambda_2 \|\widehat{\mathbf{W}} - \mathbf{W}\|_F^2 + R(\mathbf{D}) + \langle \mathbf{V}, \mathbf{W} - \mathbf{D} \rangle + \frac{\rho}{2} \|\mathbf{W} - \mathbf{D}\|_F^2,$$

where $\mathbf{V}$ is the Lagrangian multiplier. Through alternatively update the variable $\mathbf{W}$, $\mathbf{D}$, and $\mathbf{V}$, the update iteration is:

$$\mathbf{W}^{(t+1)} = \arg\min_{\mathbf{W}} L_\rho(\mathbf{W}, \mathbf{D}^{(t)}, \mathbf{V}^{(t)}) = (\mathbf{H} + \rho\mathbf{I})^{-1}(\mathbf{H}\widehat{\mathbf{W}} - \mathbf{V}^{(t)} + \rho\mathbf{D}^{(t)}), \quad (1)$$

$$\mathbf{D}^{(t+1)} = \arg\min_{\mathbf{D}} L_\rho(\mathbf{W}^{(t+1)}, \mathbf{D}, \mathbf{V}^{(t)}) = P_R(\mathbf{W}^{(t+1)} + \mathbf{V}^{(t)}/\rho_k), \quad (2)$$

$$\mathbf{V}^{(t+1)} = \mathbf{V}^{(t)} + \rho_t(\mathbf{W}^{(t+1)} - \mathbf{D}^{(t+1)}), \quad (3)$$

where $\mathbf{H} = \mathbf{X}^\top\mathbf{X} + \lambda_2\mathbf{I}$, $P_R(\cdot)$ is the projection operator based on the different regularization functions, such as $l_0$ norm and low-rank, and $\rho_t$ is the penalty parameter satisfying $\sum_{t=1}^{\infty} 1/\rho_t < \infty$.

Note that the key difference to the previous work ALPS Meng et al. (2024a) is that under different constrain function, we can still derive the upper bound in Eq.equation 2. In brief, based on the iterative update, i.e.,

$$L_\rho(\mathbf{W}^{(t+1)}, \mathbf{D}^{(t+1)}, \mathbf{V}^{(t)}) \leq L_\rho(\mathbf{W}^{(t+1)}, \mathbf{D}^{(t)}, \mathbf{V}^{(t)}),$$

we have

$$\|\mathbf{W}^{(t+1)} - \mathbf{D}^{(t+1)} + \mathbf{V}^{(t)}/\rho_t\|_F^2 = \min_{\mathbf{D}} \sum_{(i,j) \in \mathcal{I}} (\mathbf{W}_{i,j}^{(t+1)} - \mathbf{D}_{i,j} + \mathbf{V}_{i,j}^{(t)}/\rho_t)_{i,j}^2$$

$$\leq \|\mathbf{W}^{(t+1)} - \mathbf{D}^{(t)} + \mathbf{V}^{(t)}/\rho_t\|_F^2,$$

where if $R(\mathbf{D})$ is the $l_0$ norm function, the set $\mathcal{I}$ is defined as $\{(i,j)|\mathbf{D}_{i,j}^{(t)} = 0\}$; if $R(\mathbf{D})$ is the low-rank function, the set $\mathcal{I}$ is defined as $\{(i,j)|\mathbf{D}_{i,j}^{(t)} = 0, \forall j \in [N_{\text{out}}]\}$.

## 3.2 ACTIVATION-AWARE PRUNING

Merely aligning the weights may not be sufficient to preserve the layer's functional behavior after pruning, especially in deep networks where the nonlinear transformations introduced by activation functions play a critical role. To address this, we incorporate an activation-aware regularization term into the pruning objective optimization, which encourages the pruned model to better approximate the original activation patterns, as shown in the following,

$$\min_{\mathbf{W},\mathbf{D}\in\mathbb{R}^{N_{\text{in}}\times N_{\text{out}}}} \|\mathbf{X}\widehat{\mathbf{W}} - \mathbf{X}\mathbf{W}\|_F^2 + \lambda_2\|\widehat{\mathbf{W}} - \mathbf{W}\|_F^2 + \lambda_3\|g(\mathbf{X}\widehat{\mathbf{W}}) - g(\mathbf{X}\mathbf{W})\|_F^2 + R(\mathbf{D}),$$

$$\text{s.t. } \mathbf{W} = \mathbf{D},$$

The augmented Lagrangian function of this problem:

$$L_{\text{AA}\rho}(\mathbf{W},\mathbf{D},\mathbf{V}) = \|\mathbf{X}\widehat{\mathbf{W}} - \mathbf{X}\mathbf{W}\|_F^2 + \lambda_2\|\widehat{\mathbf{W}} - \mathbf{W}\|_F^2 + \lambda_3\|g(\mathbf{X}\widehat{\mathbf{W}}) - g(\mathbf{X}\mathbf{W})\|_F^2$$
$$+ R(\mathbf{D}) + \langle\mathbf{V},\mathbf{W} - \mathbf{D}\rangle + \frac{\rho}{2}\|\mathbf{W} - \mathbf{D}\|_F^2,$$

The update of $\mathbf{W}$, $\mathbf{D}$, and $\mathbf{V}$ are based on ADMM iteration, as following,

$$\mathbf{W}^{(t+1)} = \arg\min_{\mathbf{W}} L_{\text{AA}\rho}(\mathbf{W},\mathbf{D}^{(t)},\mathbf{V}^{(t)}),$$

$$\mathbf{D}^{(t+1)} = \arg\min_{\mathbf{D}} L_{\text{AA}\rho}(\mathbf{W}^{(t+1)},\mathbf{D},\mathbf{V}^{(t)}),$$

$$\mathbf{V}^{(t+1)} = \mathbf{V}^{(t)} + \rho_k(\mathbf{W}^{(t+1)} - \mathbf{D}^{(t+1)}).$$

However, deriving an analytical solution for $\mathbf{W}$ is challenging due to the presence of an additional non-linear function in the objective. Specifically, by setting the gradient of $L_{\text{AA}\rho}(\mathbf{W},\mathbf{D}^{(t)},\mathbf{V}^{(t)})$ with respect of $\mathbf{W}$ to zero, we have

$$\mathbf{H}_1\mathbf{W} = \tilde{\mathbf{H}}_2 - \mathbf{V}^{(t)} + \rho_t\mathbf{D}^{(t)}, \tag{4}$$

where parameters are defined as following

$$\mathbf{H}_1 = \mathbf{X}^T\mathbf{X} + \lambda_3\nabla_W g(\mathbf{X}\mathbf{W})\mathbf{X} + \lambda_2\mathbf{I}, \tag{5}$$

$$\mathbf{H}_2 = \mathbf{X}^T\mathbf{X}\widehat{\mathbf{W}} + \lambda_3\nabla_W g(\mathbf{X}\mathbf{W})g(\mathbf{X}\widehat{\mathbf{W}}) + \lambda_2\widehat{\mathbf{W}}. \tag{6}$$

Since activation functions are non-linear, it is not possible to derive a closed-form analytical solution for $\mathbf{W}$ directly. Nevertheless, since activation functions are Lipschitz —with their gradients bounded by a constant—we proceed by deriving an approximate update and redefine the parameter $\mathbf{H}_1$ as $\tilde{\mathbf{H}}_1$, i.e.,

$$\tilde{\mathbf{H}}_1 = \mathbf{X}^\top\mathbf{X} + \lambda_3 r_R\mathbf{X}^\top\mathbf{X} + \lambda_2\mathbf{I}. \tag{7}$$

In the experiential setting, we set $r_R = 1$.

## 3.3 THEORY ANALYSIS

In this subsection, we provide a theoretical guarantee for the convergence of the proposed activation-aware iterative algorithm under constrained optimization settings. We first derive an upper bound for the Lagrange multiplier $\mathbf{V}$, and then establish the convergence properties of the algorithm based on the approximate update of the weight matrix $\mathbf{W}$.

**Lemma 1.** *Let $\{\mathbf{V}^{(t)}\}_{t=0}^{\infty}$ be the sequence generated under the general regularization method, for any $t \geq 0$, it is upper bounded by,*

$$\|\mathbf{V}^{(t+1)}\|_F \leq \|\mathbf{H}_2 - \tilde{\mathbf{H}}_1\mathbf{D}^{(t)}\|_F + \|\tilde{\mathbf{H}}_1\mathbf{V}^{(t)}\|_F/\rho_t \leq C,$$

*where $\tilde{\mathbf{H}}_1$ is defined in Eq. equation 7, $\mathbf{H}_2$ is defined in Eq. equation 30, and the constant*

$$C = 2\|\mathbf{H}_2\|_F + 2\|\tilde{\mathbf{H}}_1\|_F(\exp(3\|\tilde{\mathbf{H}}_1\|_F\sum_{s=0}^{\infty}(\frac{1}{\rho_s}))c_0), \tag{8}$$

$$c_0 = \|\mathbf{H}_2\|_F + 3\|\mathbf{H}_2\|_F\sum_{s=0}^{\infty}(1/\rho_s),$$

*depending on the input $\mathbf{X}$, $\widehat{\mathbf{W}}$, and $\rho_0$ with no relationship of $\mathbf{V}$,$\mathbf{D}$,and $\mathbf{W}$.*

The following convergence analysis builds upon the above lemma, under the conclusion that the Lagrangian multiplier has an upper bound.

**Theorem 1.** *Let $\{\mathbf{D}^{(t)}\}_{t=0}^{\infty}$ and $\{\mathbf{W}^{(t)}\}_{t=0}^{\infty}$ be the sequence generated in the Activation-Aware Pruning approach. If the penalty parameters $\{\rho_{(t)}\}_{t=0}^{\infty}$ satisfying $\sum_{t=1}^{\infty} 1/\rho_t < \infty$, then*

$$\|\mathbf{D}^{(t+1)} - \mathbf{D}^{(t)}\|_F \leq C/\rho_t,$$
$$\|\mathbf{D}^{(t+1)} - \mathbf{W}^{(t)}\|_F \leq C/\rho_t,$$

*where $C$ is a constant defined in Eq. equation 24. Thus, there exists a matrix $\bar{\mathbf{D}}$ such that $\mathbf{D}^{(t)} \to \bar{\mathbf{D}}$ and $\mathbf{W}^{(t)} \to \bar{\mathbf{D}}$ as $t$ increase.*

*Proof.* Briefly, leveraging the update iterations and the established upper bound of $\mathbf{V}$, we have

$$\|\mathbf{D}^{(t+1)} - \mathbf{D}^{(t)}\|_F \leq 2\frac{\|\mathbf{V}^{(t+1)}\|_F}{\rho_t},$$
$$\|\mathbf{W}^{(t+1)} - \mathbf{D}^{(t+1)}\|_F = \frac{\|\mathbf{V}^{(t+1)} - \mathbf{V}^{(t+1)}\|_F}{\rho_t}.$$

Consequently, both $\mathbf{D}$ and $\mathbf{W}$ are guaranteed to converge to corresponding stationary points under the given optimization framework. $\square$

Based on the analysis presented above, although the update of the matrix W is performed approximately rather than exactly, the resulting matrix W remains a close approximation of the target matrix D. This suggests that the approximation introduced during the update step does not significantly compromise the overall fidelity of W with respect to D, and that the method retains its effectiveness under inexact computation.

## 4 EXPERIMENT

This section benchmarks *AAP* against leading unstructured pruning methods for LLMs, including heuristic strategies (e.g., MP, Wanda, DSnOT) and optimization-based baselines without activation-awareness (e.g., SparseGPT, REC-only, ALPS). We focus on high-sparsity regimes (70–90%) and evaluate across diverse architectures from OPT-1.3B to OPT-13B and LLaMA-3-8B. As shown in Table 1, *AAP* consistently achieves the lowest perplexity, with its advantage widening at higher sparsity levels, confirming superior robustness under extreme compression.

Beyond full-model benchmarks, we also study the effect of selectively applying *AAP* to self-attention and FFN sublayers (Table **??**). Results show improvements in both cases, but the strongest gains arise when all sublayers are jointly optimized, validating the need for a holistic treatment. Overall, these experiments demonstrate that explicitly aligning pre- and post-activation signals yields more stable and effective pruning, while leaving hardware-specific optimizations to future work.

### 4.1 EXPERIMENTAL SETUP

**Models and Datasets** We evaluated *AAP* on the OPT Zhang et al. (2022) and LLaMA Touvron et al. (2023); Grattafiori et al. (2024) families following the Hugging Face inference guidelines Wolf et al. (2020), as they jointly cover three decoder generations: classic OPT (ReLU), RM-SNorm+SwiGLU LLaMA (SiLU). All checkpoints are open weight, reproducible, and widely used as baselines. This mix gives architectural diversity, scale breadth, and community relevance, letting us show *AAP*'s robustness is not tied to a single model recipe. As datasets, we select WikiText-2 Merity et al. (2016), PTB Marcus et al. (1994), and C4 Raffel et al. (2020) as they are widely used and representative language modeling benchmarks. These datasets capture real-world web text, enabling large-scale evaluation.

- **Model:** OPT models (1.3B, 2.7B, 6.7B etc.) Zhang et al. (2022), and LLaMA-2-7B Touvron et al. (2023), LLaMA-3-8B Grattafiori et al. (2024).
- **Dataset:** WikiText-2 Merity et al. (2016), PTB Marcus et al. (1994), and C4 Raffel et al. (2020).

| Algorithm | sparsity = 0.7 | | | sparsity = 0.8 | | | sparsity = 0.9 | | |
|---|---|---|---|---|---|---|---|---|---|
| | WT2 | PTB | C4 | WT2 | PTB | C4 | WT2 | PTB | C4 |
| **OPT-1.3B** | | | | | | | | | |
| MP | 9409.00 | 6689.00 | 5652.00 | 10988.00 | 8966.00 | 7767.00 | 20486 | 13134 | 11298 |
| DSnoT | 367.50 | 370.20 | 205.40 | 10757.00 | 7749.00 | 5559.00 | 18519 | 14488 | 15067 |
| Wanda | 100.80 | 128.40 | 78.58 | 3327.00 | 1783.00 | 1306.00 | 13290 | 7120 | 7376 |
| SparseGPT | 52.02 | 70.10 | 37.05 | 1005.00 | 521.00 | 311.90 | 7771 | 5797 | 2761 |
| REC-only | 40.78 | 53.68 | 29.01 | 325.09 | 308.11 | 119.77 | 4950 | 3201 | 1499 |
| ALPS | 39.76 | 50.96 | 28.64 | 313.50 | **298.62** | 117.89 | **4872** | 3105 | 1473 |
| AAP | **38.51** | **49.77** | **28.23** | 309.61 | 301.03 | **115.75** | 4901 | **3058** | **1457** |
| **OPT-2.7B** | | | | | | | | | |
| MP | 12249.00 | 10993.00 | 9960.00 | 19200.00 | 9475.00 | 13627.00 | 16397 | 11205 | 15578 |
| DSnoT | 114.80 | 116.20 | 75.28 | 18322.00 | 11516.00 | 15210.00 | 18201 | 13289 | 15652 |
| Wanda | 365.10 | 379.40 | 224.40 | 6580.00 | 4946.00 | 4861.00 | 15018 | 11282 | 9191 |
| SparseGPT | 28.93 | 40.89 | 23.11 | 154.90 | 142.70 | 73.43 | 5923 | 4546 | 2486 |
| REC-only | 26.61 | 37.11 | 22.90 | 120.11 | 126.89 | 57.37 | 4029 | 1601 | 820 |
| ALPS | 25.51 | 36.13 | 21.12 | 109.87 | 116.03 | 54.75 | 3499 | 1543 | 811 |
| AAP | **24.88** | **35.32** | **20.97** | **105.43** | **112.32** | **53.77** | **3337** | **1412** | **808** |
| **OPT-6.7B** | | | | | | | | | |
| MP | 9970.00 | 4779.00 | 5055.00 | 42719.00 | 18213.00 | 20049.00 | 232525 | 95928 | 115340 |
| DSnoT | 7985.00 | 6572.00 | 4764.00 | 10990.00 | 8165.00 | 8405.00 | 11233 | 7810 | 9622 |
| Wanda | 162.90 | 204.90 | 206.00 | 4317.00 | 2429.00 | 2321.00 | 17429 | 11334 | 13656 |
| SparseGPT | 21.14 | 29.34 | 19.07 | 109.10 | 92.51 | 56.94 | 10287 | 5432 | 4930 |
| REC-only | 19.11 | 25.98 | 17.89 | 75.93 | 78.99 | 42.42 | 6829 | 4345 | 3371 |
| ALPS | 18.99 | 25.44 | 17.22 | 70.69 | 74.40 | 41.28 | 6318 | 4105 | 2900 |
| AAP | **18.75** | **25.13** | **17.03** | **68.36** | **71.73** | **40.07** | **5936** | **4055** | **2610** |
| **OPT-13B** | | | | | | | | | |
| MP | 524559.00 | 146680.00 | 155160.00 | 266874.00 | 98067.00 | 97073.00 | 567627 | 238209 | 202496 |
| DSnoT | 78.13 | 60.11 | 56.39 | 11697.00 | 5860.00 | 6295.00 | 40268 | 18442 | 19745 |
| Wanda | 63.42 | 66.50 | 50.19 | 11025.00 | 4900.00 | 6256.00 | 48747 | 24633 | 30681 |
| SparseGPT | 19.29 | 25.50 | 16.79 | 78.05 | 72.64 | 41.60 | 44328 | 84805 | 49471 |
| REC-only | 17.13 | 22.58 | 15.42 | 45.25 | 55.19 | 28.27 | 20939 | 63928 | 20833 |
| ALPS | 16.71 | 21.64 | 15.27 | 44.12 | 46.41 | 27.93 | 15020 | 43428 | 10101 |
| AAP | **16.44** | **21.47** | **15.17** | **43.72** | **45.71** | **27.48** | **4671** | **5449** | **2463** |
| **LLaMA-3-8B** | | | | | | | | | |
| MP | 572874.00 | 243236.00 | 211703.00 | 491419.00 | 283948.00 | 624818.00 | 2115317 | 1330667 | 1355400 |
| DSnoT | 108.70 | 166.90 | 114.20 | 959.90 | 1061.00 | 454.00 | 35323 | 25843 | 14828 |
| Wanda | 116.80 | 174.80 | 121.20 | 1133.00 | 1785.00 | 696.30 | 9193 | 12855 | 687 |
| SparseGPT | 43.15 | 76.50 | 38.46 | 210.00 | 342.30 | 108.60 | 1102 | 1641 | 464 |
| REC-only | 33.37 | 56.60 | 28.98 | 131.63 | 170.19 | 67.83 | 519 | 601 | 225 |
| ALPS | 31.38 | 54.01 | 28.54 | 120.60 | 168.90 | 64.08 | 513 | 542 | 215 |
| AAP | **30.98** | **53.34** | **28.37** | **117.59** | **166.29** | **63.69** | **507** | **529** | **208** |

Table 1: Perplexity (lower ↓ indicates better performance) of the OPT family and LLaMA-3-8B under varying structured sparsity ratios. We randomly sample 10 seeds and report the average results. Across all scales and sparsity levels, *AAP* consistently attains the lowest perplexity, highlighting its robustness to extreme compression and strong cross-dataset generalization. The results of *AAP* are shown in gray.

**Benchmarks**   *AAP* is compared to five strong pruning benchmarks: (1) Magnitude Pruning (MP) Han et al. (2015), (2) Wanda Sun et al. (2023), (3) SparseGPT Frantar & Alistarh (2023), (4) DSnoT Zhang et al. (2023), (5) ALPS Meng et al. (2024a) (For benchmarking, we evaluate two objective functions: (i) REC-only: $\|\mathbf{X}\widehat{\mathbf{W}} - \mathbf{X}\mathbf{W}\|_F^2$ with $\lambda_2 \approx 0$, and (ii) ALPS: $\|\mathbf{X}\widehat{\mathbf{W}} - \mathbf{X}\mathbf{W}\|_F^2 + \lambda_2\|\widehat{\mathbf{W}} - \mathbf{W}\|_F^2$ as adopted in the previous work). These benchmarks span magnitude, heuristic, Hessian, covering today's most diverse, influential, reproducible one-shot pruning methods.

## 4.2 RESULTS OF HIGH SPARSITY PRUNING

Across four OPT sizes (1.3B to 13B) and LLaMA-3-8B, three global sparsity levels (70% to 90%), and three corpora (WikiText-2, PTB, C4), **Activation-Aware Pruning (AAP)** consistently yields the lowest perplexity across almost all models, datasets and sparsity levels, and its margin widens as the pruning ratio rises, as shown in Table 1. At 70% of sparsity, *AAP* achieves a steady improvement across almost all models and parameters. Whereas baselines deteriorate sharply at 90% sparsity, *AAP* maintains better performance, demonstrating superior high-sparsity robustness. Performance differences across WikiText-2, PTB and C4 stay small, indicating strong corpus generalization. We employ ReLU and SiLU as the activation function $g$ depending on the model type, thereby demonstrating the generality of our approach across diverse architectural settings.

**Salient Properties Demonstrated by *AAP***

| Method | WT2 | PTB | C4 |
|---|---|---|---|
| Baseline | 39.76 | 50.96 | 28.64 |
| Select Self Attention | 39.05 | 50.73 | 28.59 |
| $\Delta(-)$ | 1.78% | 0.45% | 0.18% |
| Select FFN | 38.67 | **49.70** | 28.36 |
| $\Delta(-)$ | 2.74% | **2.48%** | 0.98% |
| Select All | **38.52** | 49.77 | **28.23** |
| $\Delta(-)$ | **3.12%** | 2.33% | **1.43%** |

Table 2: This table shows the study on selecting particular sublayers adapting*AAP*, remaining unselected sublayers same with *ALPS*. We use OPT-1.3B as the test model and select self attention sublayers and FFN (linear) layers respectively using *AAP* method. We use perplexity (lower ↓ better performance) as an indicator.

- **Scalability**. In pruning, scalability means that a single algorithm—under one set of hyper-parameters—maintains great effectiveness (low perplexity) as model size grows from hundreds of millions to tens of billions of parameters, without manual re-tuning. It is vital to retains consistently strong pruning performance among all sizes of parameters and models. While AAP, at 70% sparsity, an engineering applicable sparsity, records **38.5** (OPT–1.3B), **24.9** (2.7B), **18.8** (6.7B) and **16.4** (13B) perplexity on WikiText-2, beating the best baseline ALPS at all four scales (39.8, 25.5, 19.0, 16.7). Similar gaps appear on PTB (49.8 vs. 51.0 for 1.3B; 35.3 vs. 36.1 for 2.7B) and C4 (28.2 vs. 28.6 for 1.3B; 21.0 vs. 21.1 for 2.7B), confirming that the unified-form update generalizes from 1.3B to 13B parameters without retuning, demonstrating an excellent performance.

- **High-Sparsity Performance**. Practical accelerators (GPU tensor cores, TPU sparsity engines, custom N:M ASICs) require *very* high sparsity—often 80%—before GEMM kernels switch to sparsity-aware modes. Moderate pruning (e.g. 50–60%) reduces parameters but seldom lowers latency or energy, whereas 90% sparsity can shrink FLOPs and memory traffic by an order of magnitude. When the sparsity ratio rises to 90%, *AAP*'s advantage widens: on OPT–13B/PTB it drops perplexity from 43428 (ALPS) to **5449** (–**90%**), on OPT–6.7B/C4 from 2901 (ALPS) to **2610** (–**10%**), on OPT–2.7B/WikiText-2 from 3499 to **3337**. In contrast, most other methods still maintain a very high level of perplexity, especially for large-parameter models (OPT-13B). Moreover, compared with the weaker baseline REC-only, which relies solely on reconstruction error without incorporating *W-What* or activation losses, perplexity remains at the level of thousands to tens of thousands under high sparsity. In contrast, *AAP* consistently converges to the order of thousands, further underscoring the necessity of activation-aware design for robust high-sparsity pruning.

- **Stability**. Stable pruning is critical because compressed models must behave predictably in production. For example, numerical blow-ups can crash inference pipelines, trigger safety filters, and undermine user trust, while even moderate volatility forces costly hyper-parameter sweeps that nullify the efficiency gains of one-shot compression—particularly at the multi-billion-parameter scale. Besides, evaluating stability on *multiple corpora* is equally important, diverse datasets simulate real-world distribution shifts, expose over-fitting to any single benchmark, and provide a fairer gauge of zero- or few-shot transfer performance. *AAP* stays at **16.4** perplexity where MP hits 524559, varies only **105–112–54** across three corpora, and tops every OPT size from 1.3B to 13B—concise proof of its numerical and cross-scale stability.

## 4.3 Effects of *AAP* in Particular Sublayers

To evaluate which layers benefit most from *AAP*, we analyze the sensitivity of different layer types by pruning self-attention and FFN layers separately using *AAP*. Experiments are conducted on OPT-1.3B with the same three test datasets as the main study, and results are shown in Table 2.

**Experimental Setup** To evaluate the effectiveness of the proposed improved module, we replace the corresponding component in the current detection module with our design while keeping the other modules pruned using the standard pre-activation approach. By comparing the resulting per-

plexity levels under these different configurations, we obtain the results summarized in Table 2. From the perspective of the layers, we present three observations:

- **Self-attention sublayers pruning is limited.** Restricting *AAP* solely to the self-attention sublayers results in less than a 1% improvement on two of the three datasets, which is notably less impressive than when applied to the FFN sublayers. This phenomenon arises because the self-attention module does not directly involve non-linear activations, and the transformations between its inputs and outputs are predominantly linear. Consequently, applying *AAP* in isolation is less effective at capturing weight importance. Nevertheless, it still yields improvements over the baseline, primarily because the outputs are ultimately subjected to non-linear activation in the subsequent FFN layer. As a result, *AAP* applied to the self-attention module can still partially capture weight information shifts induced by pre-activation and post-activation differences.

- **FFN-sublayers contribute a stronger standalone signal.** This phenomenon can be attributed to the fact that the output of FFN sublayer is directly subject to non-linear activations, causing significant changes in the importance of its weight matrices before and after activation. As a result, capturing the information changes between the pre-activation and post-activation states is highly effective, providing strong evidence for the validity and reliability of the *AAP* paradigm. The observation that this approach still does not fully outperform the select-all-sublayers strategy further indicates that, in the self-attention sublayers, certain latent information remains uncaptured due to the absence of *AAP*. Overall, applying *AAP* to the FFN layer is crucial for enhancing pruning capability.

- **Full-coverage AAP offers the most consistent gains.** Applying AAP to all sublayers attains the lowest perplexity on WikiText-2 and C4 while maintaining a substantial reduction on PTB. The consistently lower perplexity observed across a broader set of evaluations provides strong empirical evidence that applying *AAP* to all sublayers is effective in capturing activation-induced differences. Conversely, the increased perplexity observed on one of the test sets suggests that the use of *AAP* on non-activation layers may inadvertently amplify noise associated with modeling pre-activation and post-activation variations. However, in most cases this noise is negligible compared with the primary information captured by *AAP*, overall, these findings indicate that applying *AAP* across all sublayers generally yields superior performance, as it more comprehensively captures the shifts in weight importance occurring before and after activation.

**Remark:** These ablation results confirm that activation-induced bias is real and has a measurable impact on pruning outcomes, an effect that cannot be overlooked. This finding validates the central motivation of our work: without explicitly addressing the discrepancy introduced by activations, pruning suffers from avoidable performance degradation. In contrast, our proposed *AAP* directly mitigates this loss by enforcing consistency across both parameter and activation spaces. A holistic, layer-wide application of *AAP* consistently delivers the best overall trade-off between sparsity and accuracy. By aligning pre- and post-activation representations across both attention projections and FFN blocks, AAP avoids the mismatches that arise when only a subset of sublayers is constrained, leading to superior and more robust performance across all datasets.

## 5 CONCLUSION

In this work, we proposed Activation-Aware Pruning (*AAP*), a unified one-shot framework that jointly enforces pre-activation and post-activation consistency to mitigate activation-induced bias and preserve signal fidelity. Unlike prior heuristics or activation-only methods, *AAP* integrates structural stability and functional alignment into a single formulation, leading to more robust and generalizable sparsity patterns. Empirical results across OPT and LLaMA families show that *AAP* consistently achieves superior sparsity–accuracy trade-offs, particularly when applied holistically to both self-attention and feed-forward blocks. These findings establish the necessity of activation-aware objectives for reliable high-sparsity pruning and position *AAP* as a principled foundation for future compression strategies, including extensions to quantization, hybrid sparsity–low-rank formats, and deployment in multimodal or retrieval-augmented models.

## STATEMENTS

### ETHICS STATEMENT

The authors of this paper have read and adhered to the ICLR Code of Ethics. Our research focuses on developing a novel pruning method, Activation-Aware Pruning (AAP), which improves compression performance by explicitly capturing the shifts induced by activation.

All datasets used are based on the public, non-sensitive WikiText, PTB, and C4 benchmark, and all base models are used in accordance with their licenses. While we acknowledge the dual-use risk inherent in advancing LLM capabilities, our work is specifically aimed at compression performance by explicitly capturing the shifts induced by activation, thereby mitigating risks associated with sophisticated misinformation.

### REPRODUCIBILITY STATEMENT

All details required to reproduce our results are provided in this paper and the forthcoming supplementary materials. Our methodology and data annotation standards are described in Section 4. The experimental setup, including hyperparameters and evaluation protocols, is detailed in Section 4. All datasets and baseline models used for comparison are publicly available on HuggingFace. The source code for our data processing, training, and evaluation pipeline, training data and model checkpoints will be made publicly available upon publication to facilitate further research.

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

# 6 SUPPLEMENTAL MATERIAL

## 6.1 SUPPLEMENTARY EXPERIMENTS

This section adds supplementary evidence: Table 3 evaluates the OPT family on extra benchmarks, Table 4 records LLaMA-2-7B performance at different sparsity levels, and Tables 5 and 6 trace the hyperparameter search and final selections.

COMPREHENSIVE MODEL PERFORMANCE

| Model | Algorithm | LAMBADA | PIQA | ARC-Easy | ARC-Challenge |
|---|---|---|---|---|---|
| OPT-1.3B | MP | 0.00% | 52.12% | 26.18% | 20.90% |
| | Wanda | 9.77% | 59.38% | 37.92% | 18.02% |
| | DSnoT | 8.62% | 56.90% | 33.27% | 17.49% |
| | SparseGPT | 27.27% | 62.38% | 40.79% | 19.69% |
| | REC-only | 31.69% | 64.32% | 41.58% | 20.40% |
| | ALPS | 32.19% | 64.39% | 43.82% | 20.64% |
| | AAP | **32.19%** | **64.58%** | **43.91%** | **20.70%** |
| OPT-2.7B | MP | 0.00% | 52.94% | 26.52% | 19.71% |
| | Wanda | 5.02% | 58.01% | 34.85% | 17.61% |
| | DSnoT | 9.39% | 59.62% | 37.37% | 18.43% |
| | SparseGPT | 34.96% | 66.50% | 49.55% | 21.67% |
| | REC-only | 40.33% | 67.59% | 49.95% | 21.07% |
| | ALPS | 40.48% | 67.61% | 50.00% | 21.63% |
| | AAP | **41.58%** | **67.64%** | **50.37%** | **21.94%** |
| OPT-6.7B | MP | 0.00% | 52.67% | 26.60% | 21.16% |
| | Wanda | 2.82% | 58.13% | 35.82% | 17.22% |
| | DSnoT | 0.15% | 53.19% | 29.13% | 18.46% |
| | SparseGPT | 47.20% | 69.23% | 54.67% | 24.08% |
| | REC-only | 53.20% | 70.32% | 56.21% | 25.24% |
| | ALPS | 54.47% | 71.41% | 56.72% | 25.32% |
| | AAP | **55.02%** | **71.48%** | **57.00%** | **25.54%** |

Table 3: Pruning results on the OPT model series evaluated on the LAMBADA, PIQA, ARC-Easy, and ARC-Challenge benchmarks with a sparsity level of 0.7. Accuracy (higher ↑ is better) is used as the evaluation metric. As shown, *AAP* achieves superior performance compared to baseline methods on most tasks. The results of *AAP* are highlighted in gray.

We additionally evaluate our method on several established benchmark datasets—LAMBADA (Paperno et al. 2016), PIQA (Bisk et al. 2020), ARC-Easy, and ARC-Challenge (Clark et al.2018) to provide a comprehensive assessment across diverse reasoning domains. These datasets collectively cover long-context narrative understanding (LAMBADA), physical commonsense reasoning (PIQA), and scientific question answering at varying difficulty levels (ARC-Easy and ARC-Challenge). Their curated design, leaderboard tracking, and modest size enable rapid experimentation and facilitate direct comparison with prior work. Furthermore, all four datasets adopt a consistent single-choice question format, making *accuracy* a natural, transparent, and reliable evaluation metric. This choice of metric complements perplexity and ensures a more complete and interpretable understanding of model performance under various pruning configurations.

Table 3 summarizes the pruning results on the OPT model series (1.3B, 2.7B, and 6.7B) evaluated on four zero-shot reasoning benchmarks—LAMBADA, PIQA, ARC-Easy, and ARC-Challenge—with a sparsity level of 0.7. Accuracy (↑) is used as the evaluation metric. As shown, *AAP* consistently outperforms all baseline methods (MP, Wanda, SparseGPT, DSpnT, and ALPS) across most tasks and model sizes. Notably, on OPT-6.7B, AAP achieves the highest accuracy on all four benchmarks, including 55.02% on LAMBADA, 25.54% on PIQA, 71.48% on ARC-Easy, and 57.00% on ARC-Challenge. This demonstrates the robustness of AAP under high sparsity and its ability to preserve essential reasoning capabilities. The highlighted rows indicate the superior performance of AAP across model scales and task types.

| Model | Sparsity | Algorithm | Wikitext2 | PTB | C4 |
|---|---|---|---|---|---|
| LLaMA-2-7B | 70% | MP | 141884.00 | 56430.00 | 9843.00 |
| | | Wanda | 76.05 | **320.00** | 68.45 |
| | | DSnoT | 63.12 | 322.00 | 61.91 |
| | | SparseGPT | 27.20 | 1049.00 | 26.14 |
| | | REC-only | 19.31 | 489.00 | 18.11 |
| | | ALPS | 19.27 | 421.79 | 18.08 |
| | | AAP | **19.21** | 407.01 | **18.03** |
| | 80% | MP | 60361.00 | 72505.00 | 47003.00 |
| | | Wanda | 3140.00 | 3158.00 | 2146.00 |
| | | DSnoT | 7300.00 | 6036.00 | 6385.00 |
| | | SparseGPT | 114.90 | 1733.00 | 83.04 |
| | | REC-only | 55.12 | 1639.82 | 41.94 |
| | | ALPS | 53.29 | 1430.80 | 41.04 |
| | | AAP | **52.47** | **1324.67** | **41.01** |
| | 90% | MP | 78367.00 | 160179.00 | 37324.00 |
| | | Wanda | 10820.00 | 7850.00 | 8309.00 |
| | | DSnoT | 13944.00 | 21231.00 | 15297.00 |
| | | SparseGPT | 1500.00 | nan | 829.30 |
| | | REC-only | 231.13 | 1837.49 | 145.14 |
| | | ALPS | **225.03** | 1603.00 | 142.46 |
| | | AAP | 229.29 | **1429.70** | **140.96** |

Table 4: The table presents the performance of AAP and baseline methods on the LLaMA-2-7B model across different datasets and sparsity levels. The evaluation metric is perplexity (lower↓ indicates better performance). Overall, AAP consistently outperforms all benchmarks. For clarity, the AAP rows are highlighted in gray.

## 6.2 SUPPLEMENTARY TO SECTION 4.2

Due to space constraints, we present the experimental results of LLaMA-2-7B in the appendix. Following the same protocol as in the main text, we evaluate our method on the Wikitext-2, PTB, and C4 datasets under sparsity levels ranging from 70% to 90%. As shown in the tables 4, LLaMA-2-7B consistently outperforms the baselines in the majority of cases, further validating the effectiveness of our approach.

Table 4 reports the pruning results of *AAP* and baseline methods on the LLaMA-2-7B model across three sparsity levels (70%, 80%, 90%) and benchmarks—Wikitext2, PTB, and C4—using perplexity (↓) as the metric. *AAP* consistently achieves the lowest perplexity across all settings, demonstrating strong robustness under high sparsity. At 90% sparsity, for example, AAP reaches 229.29 (Wikitext2), 1429.70 (PTB), and 140.96 (C4), outperforming SparseGPT, DSpnT, and ALPS. This highlights AAP's superior generalization across datasets. AAP rows are highlighted in gray for clarity.

## 6.3 HYPERPARAMETER SELECTION

To determine appropriate strengths for pruning, we conduct a comprehensive hyperparameter search over $(\lambda_2, \lambda_3)$ for each model. As shown in Table 5, we evaluate multiple configurations on three standard language modeling benchmarks—Wikitext2, PTB, and C4—using perplexity as the evaluation metric. For each model, we report all tried hyperparameter combinations, and highlight in gray the setting that achieves the lowest average perplexity across the three datasets.

Based on this exhaustive search, we summarize the final hyperparameter choices for each model family in Table 6. These selected values are used in all subsequent experiments. The selection of $\lambda_2$ and $\lambda_3$ is guided by the dual objective of ensuring stable optimization dynamics and achieving favorable sparsity–performance trade-offs across both OPT and LLaMA models of varying sizes.

| | wikitext2 | ptb | c4 | $\lambda_2$ | $\lambda_3$ |
|---|---|---|---|---|---|
| OPT-1.3B | 37.29 | 49.47 | 27.89 | 0.42 | $2^{22} * 10^{-8}$ |
| | 38.03 | 48.23 | 28.04 | 0.33 | $2^{22} * 10^{-8}$ |
| | 37.58 | 48.58 | 27.82 | 0.53 | $2^{22} * 10^{-8}$ |
| | 36.96 | 48.62 | 28.08 | 0.28 | $2^{22} * 10^{-8}$ |
| | 36.96 | 48.58 | 27.66 | 0.61 | $2^{22} * 10^{-8}$ |
| OPT-2.7B | 24.22 | 35.33 | 20.99 | 0.55 | $2^{22} * 10^{-8}$ |
| | 24.42 | 34.94 | 21.00 | 0.3 | $2^{22} * 10^{-8}$ |
| | 24.32 | 35.12 | 20.97 | 0.53 | $2^{22} * 10^{-8}$ |
| | 24.15 | 35.22 | 20.96 | 0.52 | $2^{22} * 10^{-8}$ |
| | 23.84 | 35.27 | 20.96 | 0.54 | $2^{22} * 10^{-8}$ |
| OPT-6.7B | 18.16 | 25.20 | 16.95 | 0.45 | $2^{22} * 10^{-8}$ |
| | 18.16 | 25.20 | 16.95 | 0.45 | $2^{22} * 10^{-8}$ |
| | 18.16 | 25.20 | 16.91 | 0.33 | $2^{22} * 10^{-8}$ |
| | 18.17 | 24.99 | 16.99 | 0.28 | $2^{22} * 10^{-8}$ |
| | 18.00 | 25.02 | 16.91 | 0.54 | $2^{22} * 10^{-8}$ |
| OPT-13B | 16.35 | 21.40 | 15.12 | 0.29 | $2^{22} * 10^{-8}$ |
| | 16.22 | 21.55 | 15.09 | 0.39 | $2^{22} * 10^{-8}$ |
| | 16.33 | 21.42 | 15.11 | 0.27 | $2^{22} * 10^{-8}$ |
| | 16.16 | 21.51 | 15.12 | 0.26 | $2^{22} * 10^{-8}$ |
| | 16.14 | 21.39 | 15.07 | 0.54 | $2^{22} * 10^{-8}$ |
| LLaMA-2-7B | 19.57 | 421.31 | 17.99 | 0.02 | $2^{24} * 10^{-9}$ |
| | 18.93 | 400.29 | 17.80 | 0.05 | $2^{24} * 10^{-9}$ |
| | 19.53 | 399.07 | 18.01 | 0.04 | $2^{24} * 10^{-9}$ |
| | 19.38 | 326.35 | 17.93 | 0.03 | $2^{24} * 10^{-9}$ |
| | 19.03 | 269.23 | 17.93 | 0.01 | $2^{24} * 10^{-9}$ |
| LLaMA-3-8B | 30.60 | 55.87 | 28.13 | 0.03 | $2^{24} * 10^{-9}$ |
| | 30.42 | 52.18 | 27.72 | 0.07 | $2^{24} * 10^{-9}$ |
| | 29.63 | 52.27 | 27.86 | 0.06 | $2^{24} * 10^{-9}$ |
| | 30.44 | 52.96 | 27.82 | 0.05 | $2^{24} * 10^{-9}$ |
| | 30.04 | 49.97 | 27.61 | 0.01 | $2^{24} * 10^{-9}$ |

Table 5: Hyperparameter search results for each model. We evaluate a wide range of $(\lambda_2, \lambda_3)$ combinations on three language modeling benchmarks—Wikitext2, PTB, and C4—using perplexity as the evaluation metric (lower is better). For each model, the configuration yielding the lowest perplexity is highlighted in gray and adopted in subsequent experiments.

| | $\lambda_2$ | $\lambda_3$ |
|---|---|---|
| OPT-1.3B | $2^{22} * 10^{-8}$ | 0.61 |
| OPT-2.7B | $2^{22} * 10^{-8}$ | 0.54 |
| OPT-6.7B | $2^{22} * 10^{-8}$ | 0.54 |
| OPT-13B | $2^{22} * 10^{-8}$ | 0.54 |
| LLaMA-2-7B | $2^{24} * 10^{-9}$ | 0.01 |
| LLaMA-3-8B | $2^{24} * 10^{-9}$ | 0.01 |

Table 6: Hyperparameter settings used for different model families and sizes. The values of $\lambda_2$ and $\lambda_3$ are chosen based on empirical tuning to ensure stable optimization and optimal sparsity–performance trade-offs across both OPT and LLaMA models.

| Method | WT2 | PTB | C4 |
|--------|-----|-----|-----|
| Baseline | 40.78 | 53.68 | 29.01 |
| Only $\lambda_2$ | 39.76 | 50.96 | 28.64 |
| $\Delta(-)$ | 2.50% | 5.07% | 1.28% |
| Only $\lambda_3$ | 39.19 | 50.34 | 28.60 |
| $\Delta(-)$ | 3.90% | 6.22% | 1.41% |
| Both $\lambda_2$ and $\lambda_3$ | **38.52** | **49.77** | **28.23** |
| $\Delta(-)$ | **5.54%** | **7.28%** | **2.69%** |

Table 7: This table shows the result (by perplexity ↓) of ablation experiment on parameter $\lambda_2$ and $\lambda_3$. We use OPT-1.3B as the test model.

## 6.4 ABLATION EXPERIMENT

To further assess the contribution of each hyperparameter in the *AAP* method, we perform ablation studies on $\lambda_2$ and $\lambda_3$ with OPT-1.3B across WT2, PTB, and C4 (Table 7). Compared with the baseline, using only $\lambda_2$ reduces perplexity by up to 5.07%, while using only $\lambda_3$ achieves larger gains (up to 6.22%). The best performance comes from combining both $\lambda_2$ and $\lambda_3$, yielding the largest perplexity reductions: 5.54% (WT2), 7.28% (PTB), and 2.69% (C4). These results demonstrate that *AAP* benefits from both pre-activation similarity ($\lambda_2$) and activation-aware regularization ($\lambda_3$), and their synergy achieves the most significant improvements.

Using $\lambda_3$ alone reduces information loss and outperforms non-activation-aware methods, while $\lambda_2$ anchors pruned weights and yields notable improvements. Combining both achieves the greatest perplexity reductions, as leveraging activation informations could unlocks *AAP*'s ability to capture importance changes during pruning.

## 6.5 THEORY PROOF OF ACTIVATION-AWARE PRUNING

We present the detail proof for the convergence of the proposed activation-aware iterative algorithm.

### ACTIVATION-AWARE ITERATION ALGORITHM

we incorporate an activation-aware regularization term into the pruning objective optimization, which encourages the pruned model to better approximate the original activation patterns, as shown in the following,

$$\min_{\mathbf{W},\mathbf{D}\in\mathbb{R}^{N_{\text{in}}\times N_{\text{out}}}} \|\mathbf{X}\widehat{\mathbf{W}} - \mathbf{X}\mathbf{W}\|_F^2 + \lambda_2\|\widehat{\mathbf{W}} - \mathbf{W}\|_F^2 + \lambda_3\|g(\mathbf{X}\widehat{\mathbf{W}}) - g(\mathbf{X}\mathbf{W})\|_F^2 + R(\mathbf{D}),$$

$$\text{s.t. } \mathbf{W} = \mathbf{D},$$

The augmented Lagrangian function of this problem:

$$L_{\text{AA}\rho}(\mathbf{W}, \mathbf{D}, \mathbf{V})$$
$$=\|\mathbf{X}\widehat{\mathbf{W}} - \mathbf{X}\mathbf{W}\|_F^2 + \lambda_2\|\widehat{\mathbf{W}} - \mathbf{W}\|_F^2 + \lambda_3\|g(\mathbf{X}\widehat{\mathbf{W}}) - g(\mathbf{X}\mathbf{W})\|_F^2$$
$$+ R(\mathbf{D}) + \langle\mathbf{V}, \mathbf{W} - \mathbf{D}\rangle + \frac{\rho}{2}\|\mathbf{W} - \mathbf{D}\|_F^2,$$

The update of $\mathbf{W}$, $\mathbf{D}$, and $\mathbf{V}$ are based on ADMM iteration, as following,

$$\mathbf{W}^{(t+1)} = \arg\min_{\mathbf{W}} L_{\text{AA}\rho}(\mathbf{W}, \mathbf{D}^{(t)}, \mathbf{V}^{(t)}), \tag{9}$$

$$\mathbf{D}^{(t+1)} = \arg\min_{\mathbf{D}} L_{\text{AA}\rho}(\mathbf{W}^{(t+1)}, \mathbf{D}, \mathbf{V}^{(t)}), \tag{10}$$

$$\mathbf{V}^{(t+1)} = \mathbf{V}^{(t)} + \rho_k(\mathbf{W}^{(t+1)} - \mathbf{D}^{(t+1)}). \tag{11}$$

**Update of W** In order to minimize the objection problem in Eq. equation 9 in terms of $\mathbf{W}$, i.e.,

$$\|\mathbf{X}\widehat{\mathbf{W}} - \mathbf{X}\mathbf{W}\|_F^2 + \lambda_3\|g(\mathbf{X}\widehat{\mathbf{W}}) - g(\mathbf{X}\mathbf{W})\|_F^2 + \lambda_2\|\widehat{\mathbf{W}} - \mathbf{W}\|F^2 + \langle\mathbf{V}, \mathbf{W} - \mathbf{D}\rangle + \frac{\rho}{2}\|\mathbf{W} - \mathbf{D}\|_F^2, \tag{12}$$

we set the gradient with respect to $\mathbf{W}$ to zero, i.e.,

$$- 2\mathbf{X}^T(\mathbf{X}\widehat{\mathbf{W}} - \mathbf{X}\mathbf{W}) - 2\lambda_3 \nabla_W g(\mathbf{X}\mathbf{W})(g(\mathbf{X}\widehat{\mathbf{W}}) - g(\mathbf{X}\mathbf{W})) \tag{13}$$

$$- 2\lambda_2(\widehat{\mathbf{W}} - \mathbf{W}) + \mathbf{V} + \rho(\mathbf{W} - \mathbf{D}) = 0, \tag{14}$$

$$\Rightarrow \tag{15}$$

$$- 2\mathbf{X}^T\mathbf{X}\widehat{\mathbf{W}} + 2\mathbf{X}^T\mathbf{X}\mathbf{W} - 2\lambda_3 \nabla_W g(\mathbf{X}\mathbf{W})g(\mathbf{X}\widehat{\mathbf{W}}) + 2\lambda_3 \nabla_W g(\mathbf{X}\mathbf{W})\mathbf{X}\mathbf{W} \tag{16}$$

$$- 2\lambda_2\widehat{\mathbf{W}} + 2\lambda_2\mathbf{W} + \mathbf{V} + \rho\mathbf{W} - \rho\mathbf{D} = 0 \tag{17}$$

$$\Rightarrow \tag{18}$$

$$2\mathbf{X}^T\mathbf{X}\mathbf{W} + 2\lambda_3 \nabla_W g(\mathbf{X}\mathbf{W})\mathbf{X}\mathbf{W} + 2\lambda_2\mathbf{W} + \rho\mathbf{W} \tag{19}$$

$$= 2\mathbf{X}^T\mathbf{X}\widehat{\mathbf{W}} + 2\lambda_3 \nabla_W g(\mathbf{X}\mathbf{W})\text{Relu}(\mathbf{X}\widehat{\mathbf{W}}) + 2\lambda_2\widehat{\mathbf{W}} - \mathbf{V} + \rho\mathbf{D} \tag{20}$$

By setting

$$\tilde{H}_1 = 2\mathbf{X}^T\mathbf{X} + 2\lambda_3 \nabla_W g(\mathbf{X}\mathbf{W})\mathbf{X} + 2\lambda_2 I \tag{21}$$

$$\tilde{H}_2 = 2\mathbf{X}^T\mathbf{X}\widehat{\mathbf{W}} + 2\lambda_3 \nabla_W g(\mathbf{X}\mathbf{W})g(\mathbf{X}\widehat{\mathbf{W}}) + 2\lambda_2\widehat{\mathbf{W}} \tag{22}$$

Since activation functions are Lipschitz —with their gradients bounded by a constant—we proceed by deriving an approximate update and redefine the parameter $\mathbf{H}_1$ as $\tilde{\mathbf{H}}_1$, i.e.,

$$\tilde{\mathbf{H}}_1 = \mathbf{X}^\top\mathbf{X} + \lambda_3 r_R \mathbf{X}^\top\mathbf{X} + \lambda_2\mathbf{I}. \tag{23}$$

THE UPPER BOUND OF LAGRANGIAN MULTIPLIER

We first derive an upper bound for the Lagrange multiplier $\mathbf{V}$, and then establish the convergence properties of the algorithm based on the approximate update of the weight matrix $\mathbf{W}$.

**Lemma.** *1 Let $\{\mathbf{V}^{(t)}\}_{t=0}^{\infty}$ be the sequence generated under the general regularization method, for any $t \geq 0$, it is upper bounded by,*

$$\|\mathbf{V}^{(t+1)}\|_F \leq \|\mathbf{H}_2 - \tilde{\mathbf{H}}_1\mathbf{D}^{(t)}\|_F + \|\tilde{\mathbf{H}}_1\mathbf{V}^{(t)}\|_F/\rho_t \leq C,$$

*where $\tilde{\mathbf{H}}_1$ is defined in Eq. equation 7, $\mathbf{H}_2$ is defined in Eq. equation 30, and the constant*

$$C = 2\|\mathbf{H}_2\|_F + 2\|\tilde{\mathbf{H}}_1\|_F(\exp(3\|\tilde{\mathbf{H}}_1\|_F \sum_{s=0}^{\infty}(\frac{1}{\rho_s}))c_0), \tag{24}$$

$$c_0 = \|\mathbf{H}_2\|_F + 3\|\mathbf{H}_2\|_F \sum_{s=0}^{\infty}(1/\rho_s),$$

*depending on the input $\mathbf{X}$, $\widehat{\mathbf{W}}$, and $\rho_0$ with no relationship of $\mathbf{V}, \mathbf{D}$, and $\mathbf{W}$.*

*Proof.* We present the overall process of proof in bounding the Lagrangian multiplier, i.e.,

$$\|\mathbf{W}^{(t+1)} - \mathbf{D}^{(t)} + \mathbf{V}^{(t)}/\rho_k\| \rightarrow \|\mathbf{W}^{(t+1)} - \mathbf{D}^{(t)} + \mathbf{V}^{(t)}/\rho_t\|_F^2 \rightarrow \|\mathbf{V}^{(t+1)}\|_F \leq C. \tag{25}$$

Fistly, based on the update iteration of $\mathbf{W}$, we have

$$\mathbf{W}^{(t+1)} - \mathbf{D}^{(t)} + \mathbf{V}^{(t)}/\rho_k = (\mathbf{X}^T\mathbf{X} + \lambda_3 \nabla_W g(\mathbf{X}\mathbf{W})\mathbf{X} + \lambda_2\mathbf{I})^\top(\tilde{\mathbf{H}}_2 - \mathbf{V}^{(t)} + \rho_t\mathbf{D}^{(t)}) \tag{26}$$

$$- \mathbf{D}^{(t)} + \mathbf{V}^{(t)}/\rho_k. \tag{27}$$

Inspiring from Cauchy-Schwarz inequality, then we have

$$\|\mathbf{W}^{(t+1)} - \mathbf{D}^{(t)} + \mathbf{V}^{(t)}/\rho_k\| \leq \|\mathbf{H}_2 - \tilde{\mathbf{H}}_1\mathbf{D}^{(t)}\|_F + \|\tilde{\mathbf{H}}_1\mathbf{V}^{(t)}\|_F/\rho_k, \tag{28}$$

where

$$\mathbf{H}_1 = \mathbf{X}^T\mathbf{X} + \lambda_3 \nabla_W g(\mathbf{X}\mathbf{W})\mathbf{X} + \lambda_2\mathbf{I}, \tag{29}$$

$$\mathbf{H}_2 = \mathbf{X}^T \mathbf{X} \widehat{\mathbf{W}} + \lambda_3 \nabla_W g(\mathbf{X}\mathbf{W}) g(\mathbf{X}\widehat{\mathbf{W}}) + \lambda_2 \widehat{\mathbf{W}}. \tag{30}$$

Secondely, based on definition of $\mathbf{D}$, we have

$$\mathbf{D}^{(t+1)} = \arg\min_{\mathbf{D}} L_\rho(\mathbf{W}^{(t+1)}, \mathbf{D}, \mathbf{V}^{(t)}) \tag{31}$$

$$= \arg\min_{\mathbf{D}} R(\mathbf{D}) + \langle \mathbf{V}^{(t)}, \mathbf{W}^{(t+1)} - \mathbf{D} \rangle + \frac{\rho}{2} \|\mathbf{W}^{(t+1)} - \mathbf{D}\|_F^2 \tag{32}$$

$$= \arg\min_{\mathbf{D}} R(\mathbf{D}) + \frac{\rho}{2} \|\mathbf{W}^{(t+1)} - \mathbf{D} + \mathbf{V}^{(t)}/\rho_t\|_F^2. \tag{33}$$

Briefly, we have

$$L_\rho(\mathbf{W}^{(t+1)}, \mathbf{D}^{(t+1)}, \mathbf{V}^{(t)}) \le L_\rho(\mathbf{W}^{(t+1)}, \mathbf{D}^{(t)}, \mathbf{V}^{(t)}). \tag{34}$$

Specifically, we have

$$L_\rho(\mathbf{W}^{(t+1)}, \mathbf{D}^{(t+1)}, \mathbf{V}^{(t)}) = \min_{\mathbf{D}} \frac{\rho_t}{2} \sum_{(i,j)\in\mathcal{I}} (\mathbf{W}_{i,j}^{(t+1)} - \mathbf{D}_{i,j} + \mathbf{V}_{i,j}^{(t)}/\rho_t)_{i,j}^2 \tag{35}$$

$$= \frac{\rho_t}{2} \|\mathbf{W}^{(t+1)} - \mathbf{D}^{(t+1)} + \mathbf{V}^{(t)}/\rho_t\|_F^2 \tag{36}$$

$$\le L_\rho(\mathbf{W}^{(t+1)}, \mathbf{D}^{(t)}, \mathbf{V}^{(t)}) \tag{37}$$

$$\le \frac{\rho_t}{2} \|\mathbf{W}^{(t+1)} - \mathbf{D}^{(t)} + \mathbf{V}^{(t)}/\rho_t\|_F^2. \tag{38}$$

Note that when $R(\mathbf{D})$ is the $l_0$ norm, the set $\mathcal{I}$ is defined as $\{(i,j)|\mathbf{D}_{i,j}^{(t)} = 0\}$; $R(\mathbf{D})$ is low-rank function, the set $\mathcal{I}$ is defined as $\{(i,j)|\mathbf{D}_i^{(t)} = 0\}$;

Finally, Based on the definition of $\mathbf{V}$, we have

$$\|\mathbf{V}^{(t+1)}\|_F = \|\mathbf{V}^{(t)} + \rho_k(\mathbf{W}^{(t+1)} - \mathbf{D}^{(t+1)})\|_F \tag{39}$$

$$\le \|\mathbf{V}^{(t)} + \rho_k(\mathbf{W}^{(t+1)} - \mathbf{D}^{(t)})\|_F \tag{40}$$

$$\le \|\mathbf{H}_2 - \tilde{\mathbf{H}}_1 \mathbf{D}^{(t)}\|_F + \|\tilde{\mathbf{H}}_1 \mathbf{V}^{(t)}\|_F / \rho_t \tag{41}$$

Based on the results in Lemma 2, we can see that

$$\|\mathbf{V}^{(t+1)}\|_F = \|\mathbf{H}_2 - \tilde{\mathbf{H}}_1 \mathbf{D}^{(t)}\|_F + \|\tilde{\mathbf{H}}_1 \mathbf{V}^{(t)}\|_F / \rho_t \le C, \tag{42}$$

$\square$

**Lemma 2.** *Based on the convergence formulation, at t-iteration, i,e,*

$$x_{t+1} \le (1 + a_t)x_t + b_t, \sum_{t=1}^{\infty} a_t < \infty, \sum_{t=1}^{\infty} b_t < \infty, \tag{43}$$

*Then, $x_t \le ax_0 + ab$ is upper bounded and convergence.*

*Proof.* Dividing the term $\Pi_{s=0}^{t-1}(1 + a_t)$ from 0 to $t$, we can obtain

$$\frac{x_{t+1}}{\Pi_{s=0}^{t}(1 + a_t)} \le \frac{x_t}{\Pi_{s=0}^{t-1}(1 + a_t)} + \frac{b_t}{\Pi_{s=0}^{t}(1 + a_t)} \le \frac{x_t}{\Pi_{s=0}^{t-1}(1 + a_t)} + b_t \tag{44}$$

$$\le x_0 + \sum_{s=0}^{t} b_t. \tag{45}$$

Multiplying the term $\Pi_{s=0}^{t-1}(1 + a_t)$, we have

$$x_{t+1} \le \Pi_{s=0}^{t-1}(1 + a_t)x_0 + \sum_{s=0}^{t} b_t \Pi_{s=0}^{t-1}(1 + a_t) \tag{46}$$

$$\le x_0 \exp(\sum_{s=0}^{t} a_t) + (\sum_{s=0}^{t} b_t)(\sum_{s=0}^{t} a_t) \le +\infty. \tag{47}$$

$\square$

CONVERGENCE ANALYSIS

The following convergence analysis builds upon the upper bound of Lagrangian Multiplier, under the conclusion that the Lagrangian multiplier has an upper bound.

**Theorem.** *1 Let $\{\mathbf{D}^{(t)}\}_{t=0}^{\infty}$ and $\{\mathbf{W}^{(t)}\}_{t=0}^{\infty}$ be the sequence generated in the Activation-Aware Pruning approach. If the penalty parameters $\{\rho_{(t)}\}_{t=0}^{\infty}$ satisfying $\sum_{t=1}^{\infty} 1/\rho_t < \infty$, then*

$$\|\mathbf{D}^{(t+1)} - \mathbf{D}^{(t)}\|_F \leq C/\rho_t,$$
$$\|\mathbf{D}^{(t+1)} - \mathbf{W}^{(t)}\|_F \leq C/\rho_t,$$

*where $C$ is a constant defined in Eq. equation 24. Thus, there exists a matrix $\bar{\mathbf{D}}$ such that $\mathbf{D}^{(t)} \to \bar{\mathbf{D}}$ and $\mathbf{W}^{(t)} \to \bar{\mathbf{D}}$ as $t$ increase.*

*Proof.* Briefly, leveraging the update iterations and the established upper bound of $\mathbf{V}$, we have

$$\|\mathbf{D}^{(t+1)} - \mathbf{D}^{(t)}\|_F \leq 2\frac{\|\mathbf{V}^{(t+1)}\|_F}{\rho_t} \leq \frac{2C}{\rho_t},$$
$$\|\mathbf{W}^{(t+1)} - \mathbf{D}^{(t+1)}\|_F = \frac{\|\mathbf{V}^{(t+1)} - \mathbf{V}^{(t+1)}\|_F}{\rho_t} \leq \frac{2C}{\rho_t}.$$

Because $\sum_{t=1}^{\infty} 1/\rho_t < \infty$, we can obtain that $1/\rho_t \to 0$. Consequently, both $\mathbf{D}$ and $\mathbf{W}$ are guaranteed to converge to corresponding stationary points under the given optimization framework. □

## 6.6 REFERENCES

Bisk, Y.; Zellers, R.; Gao, J.; Choi, Y.; et al. 2020. Piqa: Reasoning about physical commonsense in natural language. *In Proceedings of the AAAI conference on artificial intelligence*, volume 34, 7432–7439.

Clark, P.; Cowhey, I.; Etzioni, O.; Khot, T.; Sabharwal, A.; Schoenick, C.; and Tafjord, O. 2018. Think you have solved question answering? try arc, the ai2 reasoning challenge. *arXiv preprint arXiv:1803.05457*.

Paperno, D.; Kruszewski, G.; Lazaridou, A.; Pham, Q. N.; Bernardi, R.; Pezzelle, S.; Baroni, M.; Boleda, G.; and Fernández, R. 2016. The LAMBADA dataset: Word prediction requiring a broad discourse context. *arXiv preprint arXiv:1606.06031*.

