# OpenReview forum: "Activation‑Aware Pruning of Large Language Models"
_ICLR.cc/2026/Conference — Submitted to ICLR 2026_

### Official Review · Reviewer_FuyZ · 2025-10-22

**Soundness:** 2
**Presentation:** 3
**Contribution:** 2
**Rating:** 4
**Confidence:** 4

**Summary:**

In this paper, the authors aim to leverage both pre- and post-activation information during the pruning process of large language models (LLMs) to enhance high-sparsity pruning performance. Specifically, they introduce an activation-aware regularization term into the pruning objective to better preserve the original activation patterns. The paper also provides a theoretical analysis along with a formal proof of guarantee. Experimental results on OPT and LLaMA models demonstrate improvements over conventional approaches.

However, the concept of activation-aware pruning is not new, and the paper does not clearly show how its approach differs from prior methods. Moreover, the performance gains are relatively marginal. Overall, I lean toward a marginal rejection of this paper.

**Strengths:**

Originality:
The paper presents an attempt to integrate both pre- and post-activation information into the pruning process of LLMs. In addition, it provides a theoretical analysis along with a formal proof of guarantee.

Quality:
The technical presentation is generally sound. The formulation of the activation-aware regularization is clearly defined, and the optimization process is mathematically grounded.

Clarity:
The paper is well-structured and readable. And the derivation of the proposed objective is easy to follow.

Significance:
The peoblem this work discuss is important for LLMs pruning, namely high-sparsity pruning, which is highly relevant for efficient LLM deployment.

**Weaknesses:**

Main comments:
1. The paper aims to incorporate both pre- and post-activation information into the pruning process of large language models. However, it lacks a clear and formal definition of what constitutes pre- and post-activation information.
2. The concept of activation-aware pruning is not novel, and the paper does not provide a detailed discussion of how the proposed approach differs from prior works.
3. The authors introduce an activation-related term into the pruning objective, expressed as g(XW^)-g(XW). However, the paper does not clearly explain how this term reflects or utilizes pre- and post-activation information.
4. Under high-sparsity pruning settings, the reported improvements over existing methods are marginal and do not appear to be statistically significant.
5. The overall performance at high sparsity remains poor, suggesting that the proposed method is still far from being practically useful.
6. The experimental evaluation primarily focuses on the OPT model, which is relatively outdated. It would be more convincing to include experiments on more recent LLMs such as Qwen-2.5 or Qwen-3.

Minor comments:
1. Line 214: missing space in “ifR(D)”.
2. Line 305: “Table ??” is incomplete and should be corrected.

**Questions:**

1. Could the authors clarify what specific quantities or layers “pre-activation” and “post-activation” information refer to (e.g., linear transformation outputs vs. post-nonlinearity activations)?

2. Could the authors elaborate on how this term is derived, what intuition supports it, and whether it corresponds to minimizing a specific activation-space distance?

3. Could the authors provide a more detailed comparison between their approach and earlier activation-aware pruning methods? Highlighting key distinctions (e.g., in formulation, optimization, or theoretical justification) would strengthen the novelty claim.

4.  Could the authors conduct statistical significance tests or include standard deviations across multiple runs to substantiate the robustness of the observed gains?

5. Given that the performance at high sparsity remains low, how do the authors envision practical deployment of this method? Is the approach applicable to more recent or larger-scale models (e.g., Qwen-2.3, Qwen-3, or LLaMA-3)?

6. Please correct the missing space in “ifR(D)” (Line 214).

7. Table numbering is inconsistent or missing (Line 305, “Table ??”).

---

### Official Review · Reviewer_uNia · 2025-10-31

**Soundness:** 3
**Presentation:** 2
**Contribution:** 3
**Rating:** 4
**Confidence:** 4

**Summary:**

This paper proposes a novel pruning method for LLMs, called Activation-Aware Pruning (AAP). This method jointly minimizes the reconstruction errors between weight matrices, pre-activations, and post-activations, improving upon prior methods that only optimize for a subset of these factors. Theoretical results are included which show convergence of the method using approximate updates. Experimental results show AAP’s improved perplexity over existing methods.

**Strengths:**

The paper provides theoretical results and comprehensive experimental results, including ablations. The demonstrated results are strong and outperform existing methods. The proposed method is novel and has proven convergence.

**Weaknesses:**

The main weakness is in the presentation. Typos, grammatical errors, and lack of clarity in writing detract from the presentation. A list of some of these examples are below. Additionally, I would recommend adding Table 3 from the appendix into the main paper. A question I had after reading was regarding the downstream performance of the pruned models, so moving this table up (or at least referencing the results in the main text) would strengthen the claims of the paper.

- Formatting errors, eg on lines 305 and 410.
- Parenthetical citations are formatting incorrectly, should use citep in latex.
- Awkward phrasing, for example line 420: "AAP stays at 16.4 perplexity where MP hits 524559, varies only 105–112–54 across three corpora, and tops every OPT size from 1.3B to 13B—concise proof of its numerical and cross-scale stability.". The phrase "varies only 105–112–54 across three corpora" could be clearer. Additionally, it is well known that MP works extremely poorly for LLMs so while it is a valid baseline to include, it does not say much about the method if it outperforms MP.

**Questions:**

1. What is the complexity/cost of the pruning process relative to other methods?
2. For Table 1, if the authors have the results from 10 random seeds, could they add confidence intervals?
3. Why is Llama 2 7B in the appendix? It would strengthen the results to show results on multiple Llama models in the main text.

I am willing to raise my score if the authors are able to answer these questions satisfactorily.

---

### Official Review · Reviewer_5W6S · 2025-11-01

**Soundness:** 2
**Presentation:** 2
**Contribution:** 2
**Rating:** 2
**Confidence:** 5

**Summary:**

The paper proposes Activation-Aware Pruning (AAP) for one-shot LLM pruning. It augments standard reconstruction with a post-activation regularizer, yielding an ADMM-based update that approximately accounts for nonlinearity by a Lipschitz-motivated surrogate.

**Strengths:**

Clear motivation for bridging pre- vs post-activation mismatch in pruning.

**Weaknesses:**

1. The evaluated models are generally too small to support the paper’s claims.

2. The experimental setup is not sufficiently comprehensive. Please report the size of the calibration set and the sequence length, and broaden the settings if possible.

3. When AAP is applied to the self-attention block, there is no pointwise activation. How, then, does AAP differ from ALPS in this component—are you referring to the softmax as the relevant nonlinearity?

4. The improvements over ALPS appear incremental. At low sparsity, the performance seems essentially similar, and even at very high sparsity, the gains are modest.

Minor:

1. L182, the formula type of D

2. L224, g(.) is not defined

3. L305, ?? in table reference

**Questions:**

Please see the weakness.

---

### Official Review · Reviewer_5zXF · 2025-11-03

**Soundness:** 2
**Presentation:** 3
**Contribution:** 2
**Rating:** 4
**Confidence:** 4

**Summary:**

The paper introduces Activation-Aware Pruning (AAP), a one-shot method to compress large language models. AAP adds an activation-aware regularization term that aligns both pre-activation and post-activation outputs, aiming to reduce information loss after pruning. The optimization is derived using an ADMM-based framework with theoretical convergence guarantees. Experiments on OPT and LLaMA models show lower perplexity than baselines such as SparseGPT and ALPS, especially under high sparsity.

**Strengths:**

1. The paper presents a well-formulated activation-aware pruning framework that explicitly aligns pre- and post-activation signals, leading to stable and competitive performance even at very high sparsity levels.
2. The approach is theoretically grounded with convergence analysis and achieves effective compression without the need for any retraining or fine-tuning.
3. The paper presents a simple and clean idea, optimizing activation (pre and post activation functions) similarities in addition to weights.
4. The presentation is clear, with Figure 1 clearly illustrates what the method is.

**Weaknesses:**

1. The experimental evaluation primarily focuses on the OPT model family, which is now considered outdated and no longer representative of current large language model architectures. While the results are comprehensive within this setting, the absence of evaluations on modern models such as LLaMA-4, Qwen, or Deepseek, limits the evidence of generalization and the practical relevance of the proposed approach. As a result, it remains unclear whether the reported gains would hold on newer architectures with different normalization and activation designs.

2. The study exclusively considers unstructured pruning, without exploring structured sparsity patterns such as 2:4 or 4:8 that are directly compatible with modern hardware accelerators. Since these structured settings are critical for achieving actual inference speedups and deployment efficiency, the omission of such experiments makes it difficult to assess the real-world impact of the method beyond perplexity improvements.

3. Although the paper introduces an “activation-aware” regularization framework, the core idea is conceptually aligned with prior works such as Wanda and AWQ, which also incorporate activation information into pruning or quantization. The main distinction lies in the specific optimization formulation rather than a fundamentally new perspective. As a result, the conceptual novelty appears somewhat limited, even though the presentation and theoretical treatment are rigorous and well executed.

Part of the review is revised with LLM assistance.

**Questions:**

Please see weaknesses.

---

### Meta-Review · Area_Chair_57MT · 2025-12-26

**Summary:**

The paper proposes a one-shot unstructured pruning method for Large Language Models (LLMs) named Activation-Aware Pruning (AAP). The core idea is to introduce an activation-aware regularization term that aligns both pre-activation and post-activation outputs, solved via an ADMM-based framework with theoretical convergence guarantees. While reviewers acknowledged the sound mathematical formulation and the clear motivation (addressing the mismatch between pre/post-activation importance), there is a strong consensus that the paper fails to meet the bar for acceptance. Critical issues include the reliance on outdated benchmarks (OPT family), marginal performance gains over existing baselines (ALPS, SparseGPT), and a lack of comparison on modern architectures (e.g., Llama-3, Qwen) or structured sparsity settings.

**Reviewer Concerns:**

Not applicable as the authors did not provide any rebuttal.

**Reviewer Scores:**

The reviewers are likely to maintain or lower their initial rating as no rebuttal was provided.

---

### Decision · Program_Chairs · 2026-01-26

Reject